# Detecting Structural Change Point in ARMA Models via Neural Network Regression and LSCUSUM Methods

**DOI:** 10.3390/e25010133

**Published:** 2023-01-09

**Authors:** Xi-hame Ri, Zhanshou Chen, Yan Liang

**Affiliations:** 1School of Mathematics and Statistic, Qinghai Normal University, Xining 810008, China; 2The State Key Laboratory of Tibetan Intelligent Information Processing and Application, Xining 810008, China; 3School of Computer Science and Technology, Qinghai Normal University, Xining 810008, China; 4School of Preparatory Education for Nationalities, Ningxia University, Yinchuan 750002, China

**Keywords:** ARMA models, change point, neural networks regression, LSCUSUM test

## Abstract

This study considers the change point testing problem in autoregressive moving average (ARMA) (p,q) models through the location and scale-based cumulative sum (LSCUSUM) method combined with neural network regression (NNR). We estimated the model parameters via the NNR method based on the training sample, where a long AR model was fitted to obtain the residuals. Then, we selected the optimal model orders p and q of the ARMA models using the Akaike information criterion based on a validation set. Finally, we used the forecasting errors obtained from the selected model to construct the LSCUSUM test. Extensive simulations and their application to three real datasets show that the proposed NNR-based LSCUSUM test performs well.

## 1. Introduction

The change point problem has been extensively studied in time series analysis because time series often suffer from structural changes due to issues such as changes of governmental policy, health care quality, and critical social events. This problem has also been crucial in engineering, medicine, and economics, and numerous related articles have been published. We refer the reader to Csörgö and Horváth [1], Hušková et al. [2], Franke et al. [3], Kang and Lee [4], Lee and Chen [5], and Huh et al. [6] for a general review. 

The cumulative sum (CUSUM) test proposed by Page [7] is one of the most popular methods for change point tests due to its convenience of usage. Lee and Na [8] proposed the conventional estimate-based CUSUM test, which generally performs well, but suffers from severe size distortions and produces low powers on some occasions. Na et al. [9] and Lee [10] suggested the residual-based CUSUM test for time series models as a remedy. This test is more stable, but it suffers serious power loss when location parameters change because it can only detect the change in scale parameters. To improve upon this shortcoming, Oh [11,12] proposed a score vector-based CUSUM test and a modified residual-based CUSUM method to enhance the power performance. Lee [13] introduced a location- and scale-based CUSUM (LSCUSUM) test, and simulations confirmed its improved performance in terms of stability and power over any estimate-, residual- and score vector-based CUSUM tests developed thus far.

A key step in constructing the LSCUSUM test is to accurately estimate the residuals, i.e., fitting a correct model is important if we want to test change points in time series. The most popular time series fitting method is the classical ARMA method, which uses maximum likelihood estimation (QMLE) or least squares estimation (LSE) to fit an ARMA model. Classical linear ARMA models produce an accurate prediction when a time series truly follows them, but the prediction is not correct if the time series has obvious nonlinear characteristics. Because many time series carry nonlinear characteristics, new analytical tools such as neural network regression (NNR), support vector regression (SVR), and least absolute shrinkage and selection operator (LASSO) have been applied to fit them, and have been demonstrated to outperform classical time series models, especially when a time series has certain nonlinear and nonstationary characteristics. For surveys, we refer the reader to Hwarng and Ang [14], Hossain et al. [15], Zafar et al. [16], and Giacomazzo [17]. Harchaoui and Lévy-Leduc [18] considered the change point estimation problem as a variable selection problem and used the LASSO method to estimate multiple change points in one-dimensional piecewise constant signals. Chan et al. [19] applied a group LASSO method to estimate multiple change points in a time series. Jin et al. [20] proposed a fast algorithm to estimate multiple change points in nonstationary time series models. Qian and Su [21] studied a multiple change points estimation problem for panel data models via an adaptive group fused LASSO. For more studies about change point estimation via a penalized model selection method, please refer to Lee et al. [22], Jin et al. [23], and Ciuperca and Maciak [24], among many others.

Lee et al. [25,26] proposed an SVR with a hybridization of the CUSUM method to detect change points in ARMA and generalized autoregressive conditional heteroscedastic (GARCH) models. As is well known, the NNR can approximate the nonlinearity of a time series without knowing the underlying dynamic structure by minimizing the empirical risk. Motivated by this, we adopted NNR to obtain ARMA residuals, and then constructed the LSCUSUM statistic to test change points. Our simulations show that the proposed method outperforms methods based on SVR, the adaptive least absolute shrinkage and selection operator (ALASSO), and classical ARMA.

The rest of this paper is organized as follows. Section 2 presents the principle of the LSCUSUM test for ARMA models. Section 3 introduces the NNR method in a general framework. Section 4 proposes a forecasting method based on the NNR fitting ARMA model, herein the NNR-ARMA model, and then explains how to apply the NNR-ARMA model to the construction of the LSCUSUM test. Section 5 performs Monte Carlo simulations to evaluate the validity of the proposed method under various linear and nonlinear ARMA models. Section 6 applies the method to three real datasets for illustration. Finally, Section 7 provides concluding remarks.

## 2. LSCUSUM Test for ARMA Models

The model discussed in this paper is
(1)$$y_{t}=f(\Theta ;y_{t-1},\ldots ,y_{t-p,}\varepsilon_{t-1},\ldots ,\varepsilon_{t-q})+\varepsilon_{t},$$
where f is an unknown function to be estimated, $\Theta =(\phi_{1},\ldots ,\phi_{p},\theta_{1},\ldots ,\theta_{q},\sigma^{2})$ is an unknown parameter vector, p and q are real numbers, and $\varepsilon_{i}$ are IID random variables with mean zero, covariance $\sigma^{2}>0$, and $E{\left|\varepsilon_{t}\right|}^{4}<\infty .$ For given observations $y_{1},\ldots ,y_{n}$, we are interested in the null and alternative hypotheses
$$H_{0}:\Theta_{t}=\Theta ,\;t=1,\ldots ,n\;,$$
$$H_{1}:\Theta_{t}=\left\{\begin{array}{cc} \Theta , & t=1,\ldots ,k\\ \Theta +\Delta , & t=k+1,\ldots ,n \end{array}\right.,$$
where Θ is the unknown parameter vector, the nonzero constant vector Δ represents the parameter skip, and k=[nτ] is the location of the change point, where $\tau \in \left(0,1\right)$ and [⋅] denotes the integral function. Model (1) includes AR, ARMA, threshold ARMA (TARMA) and time-varying AR (TVAR) models as special cases.

To test the above null and alternative hypotheses, we applied the following two LSCUSUM tests proposed by Lee [13]:$${\hat{T}}_{n}^{LS}=\underset{1\leq k\leq n}{\max}\left\{\frac{1}{n{\hat{\gamma }}_{1,n}^{2}}{\left|\sum_{t=1}^{k}(y_{t}-{\hat{\varepsilon }}_{t}){\hat{\varepsilon }}_{t}-\frac{k}{n}\sum_{t=1}^{n}(y_{t}-{\hat{\varepsilon }}_{t}){\hat{\varepsilon }}_{t}\right|}^{2}+\frac{1}{n{\hat{\gamma }}_{2,n}^{2}}{\left|\sum_{t=1}^{k}{\hat{\varepsilon }}_{t}^{2}-\frac{k}{n}\sum_{t=1}^{n}{\hat{\varepsilon }}_{t}^{2}\right|}^{2}\right\},$$
$${\hat{T}}_{n}^{\max}=\underset{1\leq k\leq n}{\max}\max\left\{\frac{1}{\sqrt{n}{\hat{\gamma }}_{1,n}}\left|\sum_{t=1}^{k}(y_{t}-{\hat{\varepsilon }}_{t}){\hat{\varepsilon }}_{t}-\frac{k}{n}\sum_{t=1}^{n}(y_{t}-{\hat{\varepsilon }}_{t}){\hat{\varepsilon }}_{t}\right|,\frac{1}{\sqrt{n}{\hat{\gamma }}_{2,n}}\left|\sum_{t=1}^{k}{\hat{\varepsilon }}_{t}^{2}-\frac{k}{n}\sum_{t=1}^{n}{\hat{\varepsilon }}_{t}^{2}\right|\right\},$$
where ${\hat{\varepsilon }}_{t}=y_{t}-f(\hat{\Theta };y_{t-1},\ldots ,y_{t-p,}\varepsilon_{t-1},\ldots ,\varepsilon_{t-q})$ denotes the residuals with $\hat{\Theta }$ as the estimator of Θ, and
$${\hat{\gamma }}_{1,n}^{2}=\frac{1}{n}{\sum_{t=1}^{n}{\left(y_{t}-{\hat{\varepsilon }}_{t}\right)}^{2}{\hat{\varepsilon }}_{t}^{2}-\left(\frac{1}{n}\sum_{t=1}^{n}(y_{t}-{\hat{\varepsilon }}_{t}){\hat{\varepsilon }}_{t}\right)}^{2},$$
$${\hat{\gamma }}_{2,n}^{2}=\frac{1}{n}{\sum_{t=1}^{n}{\hat{\varepsilon }}_{t}^{4}-\left(\frac{1}{n}\sum_{t=1}^{n}{\hat{\varepsilon }}_{t}^{2}\right)}^{2}.$$

We rejected $H_{0}$ if ${\hat{T}}_{n}^{LS}>2.4503$ or ${\hat{T}}_{n}^{\max}>1.4596$ at the nominal 0.05 level. The critical values of these two tests were obtained though Monte Carlo simulations using two-dimensional standard Brownian motion. 

Next, we introduce the NNR method in a general framework, propose a fitting method based on NNR to the ARMA model, and explain how to apply the NNR-ARMA method when constructing the LSCUSUM test.

## 3. Neural Network Regression

A neural network (NN) is a simplified model obtained by abstracting and modeling the human brain, which is a parallel connection of a set of nodes called neurons. It can effectively solve complex regression and classification problems with a large number of correlated variables and make accurate predictions for time series.

In this study, we define an NN with an input layer, a hidden layer, and an output layer. We consider a feedforward net with n input nodes, one layer of H hidden nodes, and one output node. As shown in Figure 1, n nodes representing independent variables on the left form the input layer, H nodes in the middle form the hidden layer, and the rightmost node belongs to the output layer representing dependent variables. These nodes are connected according to the arrows, and the number next to an arrow marks its weight.

Considering an input vector $x=(x_{1},\ldots ,x_{n})\in {ℜ}^{n},$ the output $z_{h}(x;\theta )$ of the h-th hidden node is
$$z_{h}(x;\theta )=\psi \left(\sum_{i=1}^{n}\omega_{ih}x_{i}+\omega_{0h}\right).$$

The output of the net is
$$F(x;\theta )=\psi^{*}\left(\sum_{h=1}^{H}\upsilon_{h}z_{h}(x;\theta )+\upsilon_{0}\right)\;=\psi^{*}\left\{\sum_{h=1}^{H}\upsilon_{h}\left[\psi \left(\sum_{i=1}^{n}\omega_{ih}x_{i}+\omega_{0h}\right)\right]+\upsilon_{0}\right\},$$
where $\omega_{ih}(i=0,\ldots ,n;h=1,\ldots ,H)$ is the weight connecting the i-th input layer node and h-th hidden layer node; $\upsilon_{h}(h=0,\ldots ,H)$ is the weight of the h-th hidden layer node to the j-th dependent variable; and θ stands for all parameters $\upsilon_{0},\ldots ,\upsilon_{H}$ and $\omega_{ih}(i=0,\ldots ,n;$h=1,…,H) of the network. We also write $\upsilon ={\left(\upsilon_{0},\ldots ,\upsilon_{H}\right)}^{T}$ and $\omega =(\omega_{ih},i=0,\ldots ,n;h=1,\ldots ,$H). Here, ψ(⋅) and $\psi^{*}(\cdot )$ are activation functions of the NN, which are usually defined as S-shaped logistic functions,
$$\psi (x)=\frac{1}{1+\exp(-ax)},$$
and
$$\lim\psi (x)=\left\{\begin{array}{l} 1,x\to \infty \\ 0,x\to -\infty \end{array}\right.,$$
where a is a parameter that controls the inclination of the function. Different function types can be selected as required, such as the threshold, piecewise, Gaussian, and hyperbolic tangent functions.

The NN connection weights were adjusted through training, as follows:1.Assume that the input vector and the associated output vector are $x=x_{1},\ldots ,x_{m}\in {ℜ}^{n}$ and $y=y_{1},\ldots ,y_{m}\in ℜ$, respectively.2.Initialize the weights in the network:



$$\omega^{\left(0\right)}\;,\;\upsilon^{\left(0\right)}.$$



3.Use forward propagation to obtain the predicted value and calculate the loss function,



$$S(y,x;\theta )=\frac{1}{2}\sum_{j=1}^{m}{\left(y_{j}-F(x_{j};\theta )\right)}^{2}.$$



4.Update the connection weights:



$$\omega^{k}=\omega^{k-1}+\Delta \omega ;$$


$$\upsilon^{k}=\upsilon^{k-1}+\Delta \upsilon$$



Taking individual weights, the k-th iteration weights are
$$\omega_{{}_{ih}}^{\left(k\right)}=\omega_{ih}^{\left(k-1\right)}-\eta \left\{\frac{\partial S(y,x;\theta^{\left(k\right)})}{\partial \omega_{ih}}\right\},for\;i=0,\ldots ,n,h=1,\ldots ,H\mathrm{and}j=1,\ldots ,m.$$

Similarly,
$$\upsilon_{{}_{h}}^{\left(k\right)}=\upsilon_{h}^{\left(k-1\right)}-\eta \left\{\frac{\partial S(y,x;\theta^{\left(k\right)})}{\partial \upsilon_{h}}\right\},for\;h=1,\ldots ,H\mathrm{and}j=1,\ldots ,m.$$
where η(0<η<1) is the learning rate; k is the number of updates or iterations; and $\frac{\partial S(y,x;\theta^{\left(k\right)})}{\partial \omega_{hj}}$ and $\frac{\partial S(y,x;\theta^{\left(k\right)})}{\partial \upsilon_{h}}$ are respective error terms of the hidden unit and output unit, i.e., the gradient values or partial derivatives of the loss function with respect to them. According to the chain rule, they are:$$\frac{\partial S(y,x;\theta^{\left(k\right)})}{\partial \omega_{hj}}=\frac{\partial S}{\partial \omega_{hj}}=\frac{\partial S}{\partial F}\frac{\partial F}{\partial z}\frac{\partial z}{\partial \omega_{hj}},$$
$$\frac{\partial S(y,x;\theta^{\left(k\right)})}{\partial \upsilon_{h}}=\frac{\partial S}{\partial \upsilon_{h}}=\frac{\partial S}{\partial F}\frac{\partial F}{\partial z}\frac{\partial z}{\partial \upsilon_{h}}$$

5.Repeat steps 3 and 4 until the loss function is less than the preset threshold or the number of iterations is exhausted and the output parameter is the best parameter at present.

An NN can have multiple hidden layers, but one is generally enough. The number of nodes in the hidden layer can be large or small. Too many nodes may lead to overfitting, and too few to poor fitting. The number of neurodes *H* in the hidden layer can be determined as in Looney [27] by rule of thumb as
$$H=\left\|1.7*\log_{2}(n)\right\|+1.$$

Alternatively, one can use the Bayesian information criterion (BIC), as proposed by Swanson [28], to sequentially determine H. In this paper, the optimal number of nodes H is selected according to root mean square error (RMSE) recommendations through cross-validation.

## 4. Prediction Based on NNR-ARMA Model

Throughout this section, we assume that $\left\{y_{t}:t=1,\ldots ,n,n+1,\ldots ,n+l\right\}$ is generated according to the ARMA model (1). If the training sample is known to follow an NNR-ARMA (p,q) model with specific orders p and q, we can use this model. Otherwise, we determine the orders from the training sample as follows:

Step 1 Let $y=(y_{m},\ldots ,y_{n})$, and design the matrix
$$X={\left(X_{m},X_{m+1},\ldots ,X_{n}\right)}^{T}\;=\left(\begin{array}{cccccc} y_{m-1} & \cdots & y_{m-p} & \varepsilon_{m-1} & \cdots & \varepsilon_{m-q}\\ y_{m} & \cdots & y_{m-p+1} & \varepsilon_{m} & \cdots & \varepsilon_{m-q+1}\\ \vdots & \ddots & \vdots & \vdots & \ddots & \vdots \\ y_{n-1} & \cdots & y_{n-p} & \varepsilon_{n-1} & \cdots & \varepsilon_{n-q} \end{array}\right)$$
as the NN input. The design matrix X involves the potential innovation terms $\varepsilon_{t}(t=m-q,\ldots ,n-1).$ Thus, the long $AR(p^{\prime })$ model is first fitted to $y_{1},\ldots ,y_{n}$ to obtain the residual ${\hat{\varepsilon }}_{t}$, and then the residual ${\hat{\varepsilon }}_{t}$ is used to approximate the unobserved $\varepsilon_{t}$. See Hannan and Rissanen [29] for more details. The design matrix with $\varepsilon_{t}$ replaced is $\hat{X},$ where $m=p^{\prime }+\max(p,q)+1.$ When fitting the long $AR(p^{\prime })$ model, the order $p^{\prime }$ can be determined using the Akaike information criterion (AIC), with the maximum order set to be 10log10(n).

Step 2 Fix $p^{*}>0\;,\;q^{*}>0,$ and the underlying true ARMA orders $p\leq p^{*}$ and $q\leq q^{*},$ where $p^{*},\;q^{*}$ are known upper bounds of the true orders. For each p, q, we assume that $y_{1},\ldots ,y_{n}$ approximately satisfy
(2)$$y_{t}=f(y_{t-1},\ldots ,y_{t-p,}{\hat{\varepsilon }}_{t-1},\ldots {\hat{\varepsilon }}_{t-q})+{\hat{\varepsilon }}_{t},$$
and the NNR-ARMA (p,q) model is fitted with $y=(y_{m},\ldots ,y_{n})$ and $\hat{X},$ as in Section 3. Then, after applying the obtained NNR-ARMA (p,q) model to $y_{n+1},$ $\ldots ,y_{n+l}$, calculate the prediction errors ${\tilde{\varepsilon }}_{t}(t=1,\ldots ,l)$ and the corresponding AIC, namely, AIC=2(p+q) +lln(RSS/l), where l denotes the number of effective observations and RSS denotes the sum of squared residuals. We then select the optimal order $\left(p_{0},q_{0}\right)$ by minimizing the AICs.

Step 3 Train the NNR-ARMA $\left(p_{0},q_{0}\right)$ model for the whole training sample $y_{1},\ldots ,y_{n+l}$ and predict the testing samples with the estimated NNR-ARM $\left(p_{0},q_{0}\right)$ model. The obtained prediction errors are then used in the construction of the NNR-based LSCUSUM test ${\hat{T}}_{n}^{LS}$ and ${\hat{T}}_{n}^{\max}$.

## 5. Monte Carlo Simulation

We conducted Monte Carlo simulations to evaluate the finite sample performance of the NNR-based LSCUSUM test for linear and nonlinear ARMA models. For this, we used the AR, ARMA, TARMA, and TVAR models to generate data, and focused on the comparison of NNR-, ALASSO-, SVR-, and classical ARMA-based LSCUSUM tests. For each simulation, the empirical sizes and powers were calculated as the number of rejections of the null hypothesis of no changes out of 1000 repetitions at the 5% significance level. The sample sizes under consideration were n=300, 500, and 1000. The change point was assumed to occur at [n/2] under the alternatives. The simulations were conducted with R version 4.1.3, and we used the R packages nnet and caret for NNR, glmnet for ALASSO, and e1071 for SVR. 

**(1)** Linear ARMA model

We generated time series from the ARMA (1,1) model:$$y_{t}=\phi y_{t-1}+\theta \varepsilon_{t-1}+\varepsilon_{t},$$
where $\left|\phi \right|<1$ and $\varepsilon_{t}$ are IID normal random variables with mean zero and variance $\sigma^{2}.$ We considered the following setups for the null hypothesis:

Model 1: AR(1) with ϕ=0.3, θ=0, and $\sigma^{2}=1.$Model 2: ARMA(1,1) with ϕ=0.3, θ=0.3, and $\sigma^{2}=1.$

Under the alternative, we assumed a single parameter changed while the others remained constant. Table 1 and Table 2 report the empirical sizes and powers for the AR(1) and ARMA(1,1) models, respectively, from which it can be seen that there was no size distortion, and reasonably good powers were produced using all four methods. In particular, the NNR-based LSCUSUM test outperformed the ALASSO-, SVR-, and classical ARMA-based LSCUSUM test, and for the LSCUSUM statistic, ${\hat{T}}_{n}^{\max}$ was preferable to ${\hat{T}}_{n}^{LS}$ in terms of stability and power. In addition, we conducted simulations with a sample size of 500 and different change point locations; see Table 3. It can be seen that the empirical power is greatly affected by the position of the change point, the NNR-based LSCUSUM test has the highest power when a change occurs in the middle, and the power decreases as the change point comes nearer to the beginning and ending points of time series, indicating that the change point is more easily detected when in the middle of the sequence. 

We also found that our method is obviously superior to the classical method when the change point occurs at the front of the sequence. Although not reported here, we carried out simulations for various settings of model parameters and sample sizes, and a similar pattern was seen.

**(2)** Threshold ARMA model

We generated time series from the TARMA(1,1) model:$$y_{t}=\left\{\begin{array}{c} \phi_{1}y_{t-1}+\theta_{1}\varepsilon_{t-1}+\varepsilon_{t}.\;y_{t-1}\leq 1\\ \phi_{2}y_{t-1}+\theta_{2}\varepsilon_{t-1}+\varepsilon_{t}.\;y_{t-1}>1 \end{array}\right.$$
where $\varepsilon_{t}$ are IID normal random variables with mean zero and variance $\sigma^{2}.$ We considered the following setup for the null hypothesis:

Model 3: TARMA(1,1) with $\phi_{1}=0.1,\;\phi_{2}=-0.5,\;\theta_{1}=\theta_{2}=0.5,$ and $\sigma^{2}=1.$Model 4: TARMA(1,1) with $\phi_{1}=0.3,\;\phi_{2}=-0.5,\;\theta_{1}=\theta_{2}=0.5,$ and $\sigma^{2}=1.$

Under the alternative, we assumed a single parameter changed while the others remained constant. Table 4 and Table 5 list some empirical sizes and powers for models 3 and 4, respectively. The results show that the classical ARMA- and ALASSO-based LSCUSUM tests had severe size distortion, which became more severe as the sample size increased. The NNR- and SVR-based LSCUSUM did not have such a problem, and the NNR-based LSCUSUM test produced better powers than the SVR-based LSCUSUM test. As anticipated, the power increased as the sample size increased or the deviance between the two parameter sets before and after the change point became larger. The simulation results considering different change point positions were the same as before, i.e., the LSCUSUM test had the highest power when a change point occurred in the middle of the time series (Table 6).

**(3)** Time-varying AR model

We generated a time series from the period AR(1) (PAR) model:$$y_{t}=\phi_{t}y_{t-1}+\varepsilon_{t}$$
where $\phi_{t}$ is periodic with period t=1,2 and $\varepsilon_{t}\sim N(0,\sigma^{2}).$ We considered the following setup for the null hypothesis:

Model 5: PAR (1) with $\phi_{1}=-0.3,\;\phi_{2}=-0.5,$ and $\sigma^{2}=1.$Model 6: PAR (1) with $\phi_{1}=0.3,\;\phi_{2}=0.5,$ and $\sigma^{2}=1.$

Under the alternative, we assumed a single parameter changed while the others remained constant. Table 7 and Table 8 summarize the empirical sizes and powers, respectively, for the PAR model. The results show that no tests had size distortions, and the NNR-based LSCUSUM test outperformed the ALASSO-, SVR- and classical ARMA-based LSCUSUM tests of stability and power. As in cases 1 and 2, NNR-based ${\hat{T}}_{n}^{\max}$ was superior to the other tests. Table 9 shows a similar pattern to cases 1 and 2.

## 6. Empirical Applications

We illustrate our proposed NNR-based LSCUSUM test scheme by analyzing three sets of real data: the annual volume of discharge from the Nile River at Aswan in $10^{8}m^{3}$ from 1871 to 1970, the weekly average oil prices of the USA from 1 September 2014 to 27 August 2018, and the weekly Hang Seng Index (HSI) of Hong Kong from 4 September 2011 to 26 Jun 2022, whose respective time series were obtained from the R package TSA and the websites www.macrotrends.net (accessed on 27 June 2022) and investing.com. We used 70% samples of each dataset for training and the remaining 30% for testing. Prior to fitting the NNR-ARMA model, we inspected the autocorrelation function (ACF) and partial autocorrelation function (PACF) of each time series, looking for irregular patterns of autocorrelations, as seen in Figure 2, where plot (a) shows that the ACFs and PACFs for the Nile River dataset support stationarity to a great extent, indicating that estimation via the NNR-ARMA model with this time series would not undermine the outcomes. In contrast, plots (b) and (c) show that oil prices and HSI are nonstationary time series, and we use the log-returns of the oil prices and HIS.

We applied the NNR-based LSCUSUM test to the Nile River data and obtained ${\hat{T}}^{LS}=9.589$ and ${\hat{T}}^{\max}=2.631$, which are larger than the respective critical values 2.4503 and 1.4596 at the 0.05 level. Thus, we rejected the null hypothesis of no changes. We calculated that both statistics detected the change point at 1899, which is the conclusion of Cobb [30], MacNeill et al. [31], and Wu and Zhao [32]. They believed that there is a change (decrease) in year 1899, which may be due to the construction of a new dam at Aswan. See Figure 3, where the red vertical line indicates a change point. For comparison, we also applied the classical ARMA-based LSCUSUM test to the Nile River data. Here, we expect this method to misbehave due to the nonlinearity of the datasets (Keenan’s linearity test [33] is quite against the linearity assumption). We obtained ${\hat{T}}^{LS}=2.271<2.4503$, which does not reject the null hypothesis, whereas ${\hat{T}}^{\max}=1.468>1.4596$ rejects it, indicating that the change point occurs in 1917. The blue vertical line in Figure 3 shows that the change point for the dataset is quite away from the previously obtained one. This result demonstrates that the classical ARMA-based LSCUSUM can be severely damaged when a time series sample has a strong nonlinear feature.

Furthermore, we obtained the NNR-based ${\hat{T}}^{LS}=4.665$ and ${\hat{T}}^{\max}=2.009$ for log-returns of oil prices, which indicates the detection of change in both cases at the 0.05 level. Both statistics identified the location of a change point at 14 February 2016, whereas we found that ${\hat{T}}^{LS}=7.004$ and ${\hat{T}}^{\max}=2.153$ for log-returns of HSI, and both statistics indicate that the change point occurs on 10 December 2017. Similar to the case of oil prices, a significant change in trend was observed in the original datasets, as shown in Figure 4 and Figure 5.

## 7. Concluding Remarks

We proposed a prediction method based on the NNR-ARMA model and described how to determine an optimal model. This model was used to obtain residuals for a training time series sample, which was divided into two subseries. A long AR model was fitted to the first subseries to obtain the residuals, which were used as the error terms in the NNR-ARMA (p,q) model, and for each estimated NNR-ARMA (p,q) model, we calculated AICs based on the second subseries and selected an optimal ARMA order with the smallest AIC. Prediction errors or residuals were obtained using the determined NNR-ARMA (p,q) model and used to construct the LSCUSUM test. Monte Carlo simulations were conducted using the linear and nonlinear ARMA models with various parameter settings, including ARMA, TARMA, and TVAR. The results show that the NNR-based LSCUSUM test outperformed the classical ARMA- and ALASSO-based LSCUSUM tests, especially when the underlying model was nonlinear. Finally, our method was applied to the analysis of real datasets, namely of Nile River data, oil prices, and HSI prices, and detected one change point in all cases. Our findings support the validity of our method and its practicality in various real-world circumstances.

## Figures and Tables

**Figure 1 entropy-25-00133-f001:**
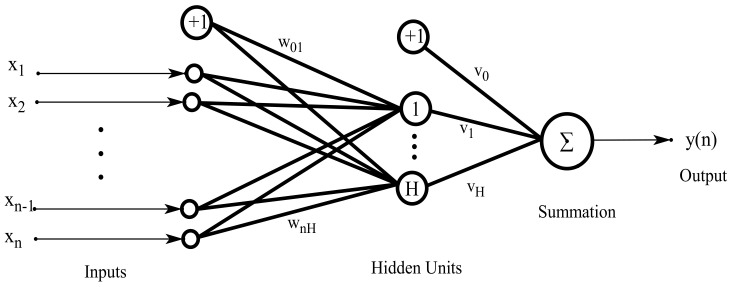
Proposed neural network topology.

**Figure 2 entropy-25-00133-f002:**
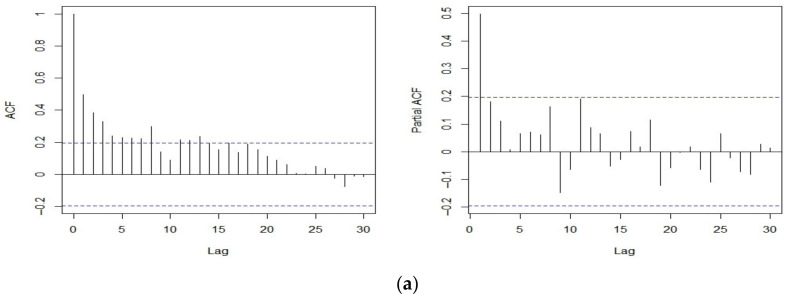
Plot of ACF and partial autocorrelation function (PACF) up to lag 30 of: (**a**) Nile River data; (**b**) oil prices; and (**c**) HSI.

**Figure 3 entropy-25-00133-f003:**
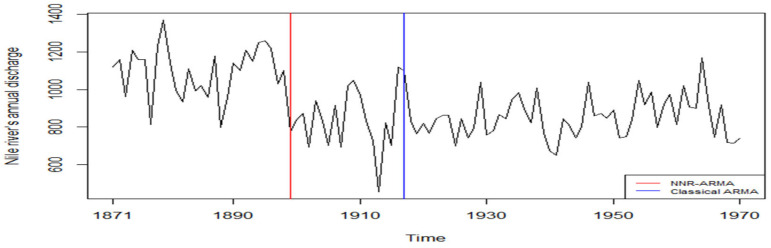
Annual volume of discharge from Nile River at Aswan, 1871–1970.

**Figure 4 entropy-25-00133-f004:**
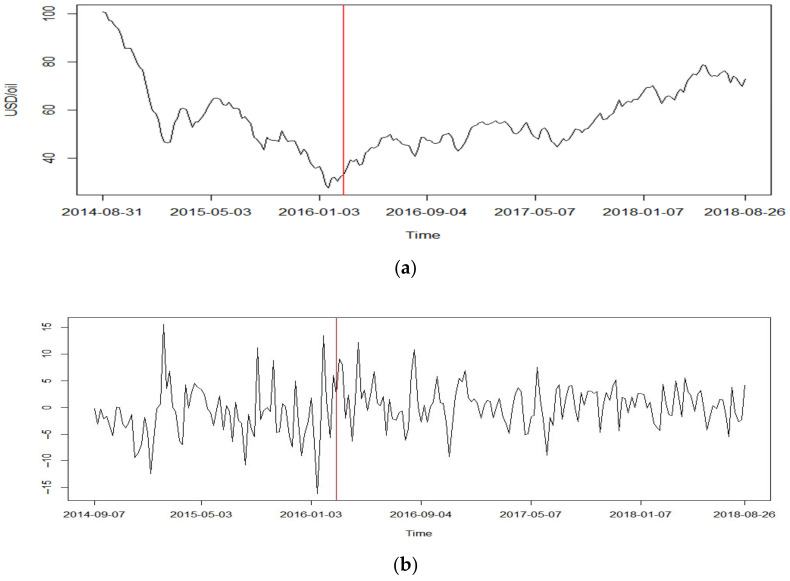
(**a**) Weekly averages of oil prices; (**b**) log-returns with detected change point.

**Figure 5 entropy-25-00133-f005:**
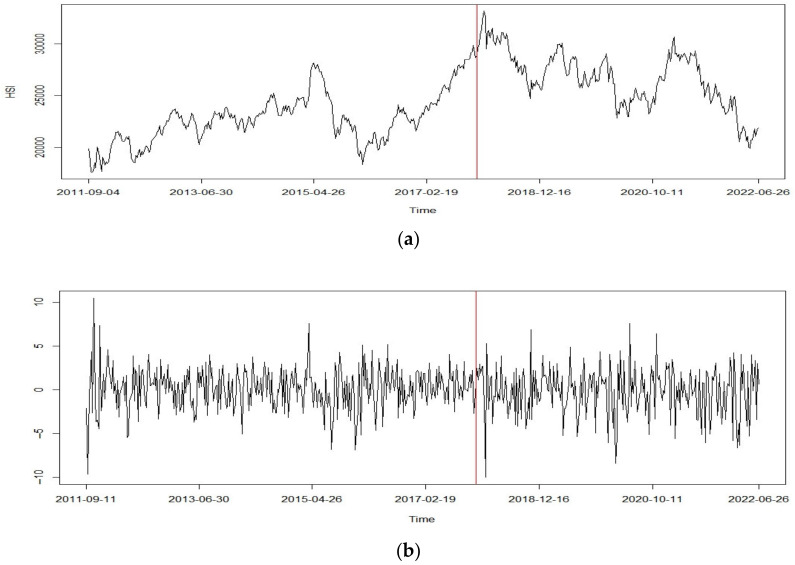
(**a**) Weekly price of HIS; (**b**) log-returns with detected change point.

**Table 1 entropy-25-00133-t001:** Empirical sizes and powers for AR(1) model with $\phi =0.3,\;\sigma^{2}=1$ under the null and each alternative.

$\phi =0.3,\sigma^{2}=1$			n=300				n=500				n=1000		
		**NNR**	**ALASSO**	**SVR**	**Classical**	**NNR**	**ALASSO**	**SVR**	**Classical**	**NNR**	**ALASSO**	**SVR**	**Classical**
Size	${\hat{T}}_{n}^{LS}$	0.056	0.044	0.038	0.043	0.046	0.045	0.047	0.046	0.061	0.044	0.044	0.045
	${\hat{T}}_{n}^{\max}$	0.050	0.042	0.035	0.041	0.045	0.040	0.046	0.042	0.057	0.039	0.043	0.039
→ϕ=0.5	${\hat{T}}_{n}^{LS}$	0.283	0.267	0.226	0.268	0.458	0.457	0.419	0.449	0.814	0.809	0.788	0.810
	${\hat{T}}_{n}^{\max}$	0.287	0.257	0.238	0.264	0.489	0.463	0.435	0.458	0.833	0.816	0.797	0.816
→ϕ=0.7	${\hat{T}}_{n}^{LS}$	0.929	0.876	0.818	0.872	0.999	0.992	0.983	0.992	1.000	1.000	1.000	1.000
	${\hat{T}}_{n}^{\max}$	0.937	0.897	0.813	0.888	0.997	0.994	0.984	0.993	1.000	1.000	1.000	1.000
$\to \sigma^{2}=2$	${\hat{T}}_{n}^{LS}$	0.908	0.903	0.885	0.916	1.000	0.999	0.997	0.999	1.000	1.000	1.000	1.000
	${\hat{T}}_{n}^{\max}$	0.930	0.923	0.990	0.934	1.000	1.000	0.998	1.000	1.000	1.000	1.000	1.000

**Table 2 entropy-25-00133-t002:** Empirical sizes and powers for ARMA(1,1) model with $\phi =0.3,\theta =0.3,\sigma^{2}=1$ under the null and each alternative.

ϕ=0.3,θ=0.3			n=300				n=500				n=1000		
$\sigma^{2}=1$		**NNR**	**ALASSO**	**SVR**	**Classical**	**NNR**	**ALASSO**	**SVR**	**Classical**	**NNR**	**ALASSO**	**SVR**	**Classical**
Size	${\hat{T}}_{n}^{LS}$	0.048	0.047	0.053	0.042	0.052	0.044	0.049	0.042	0.064	0.048	0.043	0.043
	${\hat{T}}_{n}^{\max}$	0.045	0.046	0.047	0.038	0.041	0.048	0.046	0.042	0.063	0.046	0.043	0.042
→ϕ=0.5	${\hat{T}}_{n}^{LS}$	0.316	0.281	0.254	0.252	0.512	0.473	0.420	0.439	0.863	0.830	0.790	0.805
	${\hat{T}}_{n}^{\max}$	0.316	0.279	0.251	0.242	0.522	0.474	0.408	0.440	0.870	0.850	0.794	0.824
→ϕ=0.7	${\hat{T}}_{n}^{LS}$	0.949	0.899	0.784	0.859	0.997	0.995	0.975	0.993	1.000	1.000	1.000	1.000
	${\hat{T}}_{n}^{\max}$	0.954	0.917	0.802	0.869	0.998	0.998	0.981	0.996	1.000	1.000	1.000	1.000
→θ=0.7	${\hat{T}}_{n}^{LS}$	0.488	0.515	0.451	0.492	0.798	0.829	0.764	0.813	0.992	0.994	0.984	0.988
	${\hat{T}}_{n}^{\max}$	0.519	0.527	0.451	0.512	0.810	0.837	0.774	0.821	0.993	0.993	0.984	0.992
$\to \sigma^{2}=2$	${\hat{T}}_{n}^{LS}$	0.938	0.923	0.880	0.922	0.999	0.998	0.992	0.998	1.000	1.000	1.000	1.000
	${\hat{T}}_{n}^{\max}$	0.931	0.925	0.884	0.927	0.997	0.997	0.996	0.997	1.000	1.000	1.000	1.000

**Table 3 entropy-25-00133-t003:** Empirical powers when change point occurs at [nτ] with 
n=500 for ARMA(1,1) model.

ϕ=0.3,θ=0.3			τ=0.25				τ=0.50				τ=0.75		
$\sigma^{2}=1$		**NNR**	**ALASSO**	**SVR**	**Classical**	**NNR**	**ALASSO**	**SVR**	**Classical**	**NNR**	**ALASSO**	**SVR**	**Classical**
Size	${\hat{T}}_{n}^{LS}$	0.278	0.228	0.212	0.107	0.512	0.473	0.420	0.439	0.368	0.281	0.257	0.248
	${\hat{T}}_{n}^{\max}$	0.287	0.226	0.204	0.100	0.522	0.474	0.408	0.440	0.359	0.298	0.260	0.263
→ϕ=0.5	${\hat{T}}_{n}^{LS}$	0.899	0.827	0.657	0.389	0.997	0.995	0.975	0.993	0.947	0.945	0.910	0.935
	${\hat{T}}_{n}^{\max}$	0.914	0.857	0.691	0.377	0.998	0.998	0.981	0.996	0.952	0.950	0.931	0.952
→ϕ=0.7	${\hat{T}}_{n}^{LS}$	0.600	0.676	0.509	0.375	0.798	0.829	0.764	0.813	0.551	0.494	0.451	0.458
	${\hat{T}}_{n}^{\max}$	0.628	0.684	0.527	0.389	0.810	0.837	0.774	0.821	0.541	0.497	0.450	0.452
→θ=0.7	${\hat{T}}_{n}^{LS}$	0.834	0.809	0.757	0.439	0.999	0.998	0.992	0.998	0.964	0.961	0.940	0.962
	${\hat{T}}_{n}^{\max}$	0.861	0.871	0.796	0.510	0.997	0.997	0.996	0.997	0.969	0.969	0.953	1.969

**Table 4 entropy-25-00133-t004:** Empirical sizes and powers for threshold ARMA(1,1) model with $\phi_{1}=0.1,\phi_{2}=-0.5,$ $\theta =\theta_{1}=\theta_{2}=0.5,$ $\sigma^{2}=1$ under the null and each alternative.

$\phi_{1}=0.1,\phi_{2}=-0.5$			n=300				n=500				n=1000		
$\theta =0.5,\sigma^{2}=1$		NNR	ALASSO	SVR	Classical	NNR	ALASSO	SVR	Classical	NNR	ALASSO	SVR	Classical
Size	${\hat{T}}_{n}^{LS}$	0.057	0.070	0.039	0.073	0.040	0.086	0.038	0.096	0.041	0.083	0.037	0.097
	${\hat{T}}_{n}^{\max}$	0.045	0.066	0.040	0.068	0.040	0.092	0.041	0.095	0.041	0.081	0.043	0.095
$\to \phi_{1}=0.5$	${\hat{T}}_{n}^{LS}$	0.464	0.310	0.213	0.445	0.888	0.817	0.820	0.825	1.000	1.000	0.994	1.000
	${\hat{T}}_{n}^{\max}$	0.510	0.327	0.321	0.475	0.909	0.769	0.748	0.756	1.000	1.000	0.997	1.000
$\to \phi_{1}=0.7$	${\hat{T}}_{n}^{LS}$	0.998	0.975	0.684	0.977	1.000	1.000	0.925	1.000	1.000	1.000	1.000	1.000
	${\hat{T}}_{n}^{\max}$	0.998	0.951	0.748	0.959	1.000	1.000	0.951	1.000	1.000	1.000	1.000	1.000
→θ=−0.5	${\hat{T}}_{n}^{LS}$	0.524	0.361	0.323	0.712	0.747	0.391	0.481	0.810	0.888	0.630	0.753	0.946
	${\hat{T}}_{n}^{\max}$	0.531	0.390	0.352	0.737	0.756	0.398	0.513	0.818	0.899	0.635	0.776	0.946
$\to \sigma^{2}=2$	${\hat{T}}_{n}^{LS}$	0.904	0.913	0.868	0.913	0.995	0.989	0.997	0.991	1.000	1.000	1.000	1.000
	${\hat{T}}_{n}^{\max}$	0.909	0.920	0.880	0.923	0.994	0.993	0.991	0.991	1.000	1.000	1.000	1.000

**Table 5 entropy-25-00133-t005:** Empirical sizes and powers for threshold ARMA(1,1) model with $\phi_{1}=0.3,\phi_{2}=-0.5,$ $\theta =\theta_{1}=\theta_{2}=0.5,$ $\sigma^{2}=1$ under the null and each alternative.

$\phi_{1}=0.3,\phi_{2}=-0.5$			n=300				n=500				n=1000		
$\theta =0.5,\sigma^{2}=1$		NNR	ALASSO	SVR	Classical	NNR	ALASSO	SVR	Classical	NNR	ALASSO	SVR	Classical
Size	${\hat{T}}_{n}^{LS}$	0.065	0.094	0.034	0.100	0.069	0.137	0.050	0.137	0.045	0.134	0.044	0.134
	${\hat{T}}_{n}^{\max}$	0.067	0.094	0.035	0.096	0.068	0.140	0.039	0.134	0.048	0.136	0.042	0.135
$\to \phi_{1}=0.7$	${\hat{T}}_{n}^{LS}$	0.880	0.799	0.433	0.836	0.994	0.999	0.795	1.000	1.000	1.000	0.996	1.000
	${\hat{T}}_{n}^{\max}$	0.876	0.781	0.528	0.841	0.993	0.995	0.857	0.995	1.000	1.000	0.998	1.000
$\to \phi_{1}=0.9$	${\hat{T}}_{n}^{LS}$	1.000	1.000	0.488	0.996	1.000	1.000	0.701	1.000	1.000	1.000	1.000	1.000
	${\hat{T}}_{n}^{\max}$	1.000	1.000	0.543	0.997	1.000	1.000	0.769	1.000	1.000	1.000	1.000	1.000
→θ=−0.5	${\hat{T}}_{n}^{LS}$	0.225	0.113	0.090	0.430	0.264	0.140	0.118	0.549	0.284	0.218	0.160	0.678
	${\hat{T}}_{n}^{\max}$	0.259	0.110	0.095	0.472	0.278	0.135	0.126	0.585	0.295	0.212	0.183	0.701
$\to \sigma^{2}=2$	${\hat{T}}_{n}^{LS}$	0.904	0.896	0.852	0.903	0.993	0.898	0.987	0.991	1.000	1.000	1.000	1.000
	${\hat{T}}_{n}^{\max}$	0.919	0.910	0.865	0.911	0.996	0.988	0.988	0.988	1.000	1.000	1.000	1.000

**Table 6 entropy-25-00133-t006:** Empirical power of the change point occurs at [nτ] and n=500 for threshold ARMA(1,1).

$\phi_{1}=0.1,\phi_{2}=-0.5$			τ=0.25				τ=0.50				τ=0.75		
$\theta =0.5,\sigma^{2}=1$		NNR	ALASSO	SVR	Classical	NNR	ALASSO	SVR	Classical	NNR	ALASSO	SVR	Classical
$\to \phi_{1}=0.5$	${\hat{T}}_{n}^{LS}$	0.365	0.378	0.298	0.479	0.888	0.817	0.478	0.825	0.629	0.576	0.510	0.626
	${\hat{T}}_{n}^{\max}$	0.358	0.357	0.302	0.496	0.909	0.769	0.489	0.756	0.635	0.559	0.516	0.616
$\to \phi_{1}=0.7$	${\hat{T}}_{n}^{LS}$	0.700	0.769	0.575	0.826	1.000	1.000	0.925	1.000	0.989	0.952	0.922	0.960
	${\hat{T}}_{n}^{\max}$	0.726	0.756	0.608	0.836	1.000	1.000	0.951	1.000	0.993	0.940	0.932	0.951
→θ=−0.5	${\hat{T}}_{n}^{LS}$	0.986	0.975	0.298	1.000	0.747	0.391	0.481	0.810	0.756	0.354	0.544	0.891
	${\hat{T}}_{n}^{\max}$	0.988	0.975	0.271	0.999	0.756	0.398	0.513	0.818	0.720	0.351	0.489	0.868
$\to \sigma^{2}=2$	${\hat{T}}_{n}^{LS}$	0.810	0.826	0.778	0.836	0.995	0.989	0.997	0.991	0.962	0.949	0.938	0.949
	${\hat{T}}_{n}^{\max}$	0.814	0.864	0.787	0.962	0.994	0.993	0.991	0.991	0.961	0.954	0.943	0.955

**Table 7 entropy-25-00133-t007:** Empirical sizes and powers for PAR(1) model with $\phi_{1}=-0.3,\phi_{2}=-0.5,$ T=2 under the null and each alternative.

$\phi_{1}=-0.3,\phi_{2}=-0.5$			n=300				n=500				n=1000		
T=2		NNR	ALASSO	SVR	Classical	NNR	ALASSO	SVR	Classical	NNR	ALASSO	SVR	Classical
Size	${\hat{T}}_{n}^{LS}$	0.053	0.049	0.056	0.039	0.055	0.063	0.058	0.051	0.057	0.049	0.048	0.047
	${\hat{T}}_{n}^{\max}$	0.052	0.047	0.046	0.038	0.060	0.053	0.055	0.049	0.052	0.050	0.046	0.056
$\to \phi_{1}=-0.5$	${\hat{T}}_{n}^{LS}$	0.140	0.113	0.112	0.095	0.185	0.163	0.155	0.136	0.337	0.333	0.281	0.290
	${\hat{T}}_{n}^{\max}$	0.134	0.113	0.104	0.097	0.175	0.162	0.153	0.134	0.343	0.340	0.305	0.299
$\to \phi_{1}=-0.7$	${\hat{T}}_{n}^{LS}$	0.347	0.291	0.291	0.247	0.567	0.502	0.468	0.445	0.875	0.870	0.831	0.834
	${\hat{T}}_{n}^{\max}$	0.344	0.289	0.293	0.240	0.570	0.506	0.482	0.452	0.888	0.869	0.847	0.848
$\to \phi_{2}=-0.7$	${\hat{T}}_{n}^{LS}$	0.094	0.079	0.086	0.069	0.137	0.091	0.117	0.082	0.202	0.199	0.212	0.170
	${\hat{T}}_{n}^{\max}$	0.086	0.074	0.079	0.062	0.116	0.092	0.111	0.075	0.186	0.186	0.199	0.162
$\to \phi_{2}=0.3$	${\hat{T}}_{n}^{LS}$	0.704	0.576	0.430	0.712	0.952	0.898	0.775	0.950	1.000	1.000	0.995	1.000
	${\hat{T}}_{n}^{\max}$	0.693	0.580	0.420	0.714	0.954	0.899	0.747	0.950	1.000	1.000	0.996	1.000

**Table 8 entropy-25-00133-t008:** Empirical sizes and powers for threshold PAR(1) model with $\phi_{1}=0.3,\phi_{2}=0.5,$ T=2 under the null and each alternative.

$\phi_{1}=0.3,\phi_{2}=0.5$			n=300				n=500				n=1000		
T=2		NNR	ALASSO	SVR	Classical	NNR	ALASSO	SVR	Classical	NNR	ALASSO	SVR	Classical
Size	${\hat{T}}_{n}^{LS}$	0.054	0.048	0.039	0.044	0.047	0.050	0.048	0.039	0.059	0.041	0.032	0.031
	${\hat{T}}_{n}^{\max}$	0.056	0.039	0.036	0.039	0.046	0.053	0.044	0.046	0.059	0.039	0.041	0.034
$\to \phi_{1}=0.5$	${\hat{T}}_{n}^{LS}$	0.125	0.107	0.095	0.090	0.190	0.154	0.136	0.138	0.329	0.327	0.276	0.289
	${\hat{T}}_{n}^{\max}$	0.119	0.106	0.097	0.094	0.176	0.153	0.137	0.139	0.327	0.317	0.278	0.287
$\to \phi_{1}=0.7$	${\hat{T}}_{n}^{LS}$	0.346	0.315	0.269	0.262	0.564	0.506	0.475	0.443	0.862	0.859	0.830	0.831
	${\hat{T}}_{n}^{\max}$	0.333	0.299	0.268	0.244	0.583	0.511	0.476	0.457	0.878	0.872	0.841	0.845
$\to \phi_{2}=0.7$	${\hat{T}}_{n}^{LS}$	0.111	0.081	0.087	0.065	0.140	0.086	0.100	0.083	0.237	0.178	0.196	0.140
	${\hat{T}}_{n}^{\max}$	0.098	0.074	0.075	0.062	0.125	0.086	0.097	0.070	0.230	0.160	0.182	0.128
$\to \phi_{2}=-0.3$	${\hat{T}}_{n}^{LS}$	0.717	0.589	0.417	0.754	0.953	0.921	0.777	0.961	1.000	1.000	0.987	1.000
	${\hat{T}}_{n}^{\max}$	0.722	0.580	0.413	0.738	0.950	0.924	0.767	0.964	1.000	1.000	0.987	1.000

**Table 9 entropy-25-00133-t009:** Empirical power of the change point occurs at [nτ] for PAR(1).

$\phi_{1}=-0.3,\phi_{2}=-0.5$			τ=0.25				τ=0.50				τ=0.75		
T=2		NNR	ALASSO	SVR	Classical	NNR	ALASSO	SVR	Classical	NNR	ALASSO	SVR	Classical
$\to \phi_{1}=-0.5$	${\hat{T}}_{n}^{LS}$	0.123	0.095	0.096	0.088	0.185	0.163	0.155	0.136	0.128	0.106	0.112	0.096
	${\hat{T}}_{n}^{\max}$	0.118	0.095	0.093	0.083	0.175	0.162	0.153	0.134	0.125	0.121	0.119	0.105
$\to \phi_{1}=-0.7$	${\hat{T}}_{n}^{LS}$	0.301	0.214	0.218	0.181	0.567	0.502	0.468	0.445	0.392	0.346	0.318	0.308
	${\hat{T}}_{n}^{\max}$	0.288	0.225	0.219	0.170	0.570	0.506	0.482	0.452	0.397	0.344	0.332	0.307
$\to \phi_{2}=-0.7$	${\hat{T}}_{n}^{LS}$	0.084	0.064	0.077	0.054	0.137	0.091	0.117	0.082	0.104	0.083	0.105	0.071
	${\hat{T}}_{n}^{\max}$	0.082	0.062	0.084	0.053	0.116	0.092	0.111	0.075	0.091	0.086	0.084	0.074
$\to \phi_{2}=0.3$	${\hat{T}}_{n}^{LS}$	0.496	0.318	0.270	0.608	0.952	0.898	0.775	0.950	0.792	0.751	0.613	0.746
	${\hat{T}}_{n}^{\max}$	0.512	0.315	0.270	0.621	0.954	0.899	0.747	0.950	0.798	0.763	0.600	0.744

## Data Availability

Data is contained within the article.

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
