# Peer review of "Detecting Structural Change Point in ARMA Models via Neural Network Regression and LSCUSUM Methods"

_entropy, 2023, doi:10.3390/e25010133_

Round 1

Reviewer 1 Report

The paper is clearly written and presents an interesting new methodology.

The simulation results represent a thorough exploration and justification of the method.

The 3 real data examples are welcome as a final conclusion to the paper.

The references are comprehensive.

The English language is very good. There are typos in section 6:

"[24]. They are believed that there is a change (decrease) in year 1899, " should be: "[24]. They believed that there is a change (decrease) in year 1899, "

The sentence: "See Figure 3, where red vertical lines indicate change point" should be:

"See Figure 3, where the red vertical line indicates a change point" and

"The blue vertical lines in Figure 2 show that the change..." should be:

"The blue vertical line in Figure 3 shows that the change ..."

Reviewer 2 Report

The authors consider the change point problem in autoregressive moving average models and apply the LSCUSUM and NNR to detect the change point. The topic is interesting, but could be improved. I hope my comments would be helpful for further improvement.

1. In Section 5, authors compared NNR, ALASSO, SVR-based LSCUSUM tests. The corresponding references are not cited except the R packages. Especially for ALSSSO, there are some change point papers on penalized model selection which have not been reviewed and compared.

Harchaoui, Z. & Levy-Leduc, C. (2010). Multiple change-point estimation with a total variation penalty.Journal of the American Statistical Association, 105, 1480–1493.

Jin, B., Shi, X., & Wu, Y. (2013). A novel and fast methodology for simultaneous multiple structural breakestimation and variable selection for nonstationary time series models. Statistics and Computing, 23,221–231.

Jin, B., Wu, Y., & Shi, X. (2016). Consistent two-stage multiple change-point detection in linear models. Canadian journal of statistics, 44, 161–179.

2. In page 3, two LSCUSUM tests were applied but \hat\varepsilon_t is undefined. I guess it is residual. Please confirm it and formally define it. There is also no definition of estimate of change point. The \tau is related to the change point but \hat\tau_{1,n}^2 and \hat\tau_{2,n}^2 are not related to the estimate of change pont. Please use different notations for that.

3. In page 4, the S-shaped logistic function \psi(x) should hav a limit (NOT equal) to 1 or 0 when x tends to infinity or negative infinity.

4. In page 6, the RSS appeares twice but has different fonts.

5. For better representation, the labels in Fig. 3, 4, and 5 should have normal scales.

Round 2

Reviewer 2 Report

There are some typos in references such as AStA in [9] and journal in [23].